# Anomalies, Representations, and Self-Supervision

Barry M. Dillon[1,2], Luigi Favaro[1], Friedrich Feiden[1],
Tanmoy Modak[1,3], Tilman Plehn[1]

[1] Institut für Theoretische Physik, Universität Heidelberg, Germany
[2] ISRC, Ulster University, Derry, Northern Ireland
[3] Department of Physical Sciences, Indian Institute of Science Education and Research, India

August 8, 2024

## Abstract

We develop a self-supervised method for density-based anomaly detection using contrastive learning, and test it using event-level anomaly data from CMS ADC2021. The AnomalyCLR technique is data-driven and uses augmentations of the background data to mimic non-Standard-Model events in a model-agnostic way. It uses a permutation-invariant Transformer Encoder architecture to map the objects measured in a collider event to the representation space, where the data augmentations define a representation space which is sensitive to potential anomalous features. An AutoEncoder trained on background representations then computes anomaly scores for a variety of signals in the representation space. With AnomalyCLR we find significant improvements on performance metrics for all signals when compared to the raw data baseline.

# 1   Introduction

Model-agnostic new physics searches are one of the most interesting analysis prospects for the LHC and other colliders. Over the past decade the LHC has searched for new physics based on model-specific hypothesis testing. Despite these efforts there has been no strong evidence of new physics found. It is possible that new physics does exist at the scales probed by the LHC, and has not been uncovered due to the particular signal not being covered by previous analysis hypotheses. The ATLAS and CMS collaborations have both implemented model-agnostic new physics searches to deal with this [1, 2], however these methods suffer some drawbacks. For example scanning high-dimensional parameter spaces can lead to large look-elsewhere effects, or methods can lack the ability to make full use of the high-granularity low-level information collected in the experiments. Recent progress in machine learning based high-energy physics tools are making significant advances in solving many problems of such classical methods [3].

The main machine learning tools to date for data-driven model-agnostic searches are based either on density-related scores, or on classification scores using a background-dominated control sample. The latter, typically known as CWoLa methods (Classification Without Labels) [4–6] have been shown to be very successful in applications such as bump hunting [7–14] and semi-visible jet searches [15], providing both anomaly scores and background estimates. However they run into difficulty when the dimension of the input space or number of observables becomes large, and so the question of whether or not they can be used on low level data is still uncertain. CWoLa tools have already been adopted by the ATLAS collaboration [16].

Density-based methods use machine learning to estimate the density in the phase space, and then identify anomalies as those laying in the low density regions. These tools typically work on high-dimensional inputs and so can be used on low-level data. The first density-based methods were the AutoEncoder studies [17, 18], where the network is optimised to compress and reconstruct the kinematics of a jet or event. While this is not strictly density estimation, the optimisation is highly aligned with learning a density, since regions of the phase space which are most populated are those which should be reconstructed the best and thus have the lowest anomaly score. There has been significant progress with the AutoEncoder tools and other density-based anomaly detection methods in recent years [19–33], with studies covering interpretability of AutoEncoders [34, 35], topic modelling [36, 37], null hypothesis tests for anomaly detection [38], ABCD methods [39], the Normalised AutoEncoder (NAE) [40], and normalising flow techniques [41–44]. For a comprehensive summary of many different anomaly detection methods we refer the reader to the community challenge papers in Refs [45, 46].

One issue with the density-based approaches [44, 47] is that the score is not invariant under simple transformations in the phase space. This means that a simple re-mapping of the momenta or coordinates fundamentally changes what the anomaly score is. This poses the question of how to choose a representation of the data for use in density-based anomaly detection tasks. It is also worth noting that despite the great progress that more sophisticated neural network architectures and the implementation of symmetries in networks has brought to supervised classification [48–51], they have not yet led to the same progress in anomaly detection. In this work we develop a new approach to density-based anomaly detection using self-supervision, which defines the representation of the data in a model-agnostic way using the power of highly expressive networks such as transformers or graph networks to boost anomaly detection performance.

Supervised machine learning methods use the idea of a truth-label to optimise the neural networks, usually to classify between data with different truth labels. Unsupervised methods are those which do not require truth labels, instead optimising a network using a reconstruc-

tion loss or a negative log likelihood, for example. Self-supervision on the other hand uses 'pseudo-labels', labels generated from the data without knowledge of a truth label, to optimise the networks. In contrastive learning [52], these labels correspond to a link between an original event and an augmented event. We define the augmentation as some physical modification of the event kinematics. Contrastive learning uses the pseudo-labels to devise an auxiliary task for the network optimisation through the contrastive loss function. Now the network learns how to process high-dimensional correlations in the data, and thus the representations learned by these networks can be very useful for downstream tasks. We introduced the self-supervised JetCLR method in [53] and demonstrated its ability to construct highly expressive representations for classification tasks. In [54] this same technique was used to construct representations for CWoLa-based anomaly detection. In addition to these works, other self-supervised / representation learning techniques have been applied in particle physics [55,56] and in other scientific disciplines such as astrophysics [57–60]. In [53,54] the augmentations corresponded to transformations of the event to which the underlying physics should be invariant to rotations or translations, but also soft-collinear parton splittings.

We introduce AnomalyCLR, a new method based on the idea of 'anomaly-augmentations'. These anomaly-augmentations are modifications of the original event to which the underlying physics is not invariant. In fact these augmentations are chosen to mimic very general features that anomalous events might have, such as high multiplicity, large MET, or large $p_T$. Despite choosing explicitly the augmentations, the approach does not target any specific new physics model, and we will see from the results that the approach is model agnostic. AnomalyCLR projects the kinematics of each event to a representation vector, which we then use to train an AutoEncoder and define the anomaly scores. It enriches the representation space using known invariances in the data, such as invariance to azimuthal rotations, and known generic features of anomalies. Self-supervised anomaly detection methods have gained prominence in the machine learning literature recently [61–64], and while the approaches are necessarily domain specific, we have drawn on these methods. The anomaly score can be computed in different ways [65], and we opt for the AutoEncoder approach. So the workflow is as follows; train AnomalyCLR to obtain a representation vector for each event in the dataset, then train an AutoEncoder on these representations to obtain the anomaly scores. This is in contrast to the typical approach of training the AutoEncoder directly on the raw kinematical data from the events. We test AnomalyCLR on the CMS Anomaly Detection Challenge dataset [66], and, compared to the raw data baseline, we find significant improvements on all signals.

In Section 2 we will discuss the dataset and the different backgrounds and signals. In Section 3 we will then introduce the AnomalyCLR idea, first discussing contrastive learning and then how this can be modified for use in anomaly detection. The specifics of the application to event-level collider data such as the CMS ADC dataset is given in Section 4. The discussion on how we estimate anomaly scores is given in Section 5, where the architecture and optimisation of the AutoEncoder we use is discussed. The results are presented in Section 6, along with an analysis of how different anomaly-augmentations and different representation dimensions affect the results. We conclude in Section 7 with a discussion of the results and future directions.

## 2   Dataset

To test the performance of the AnomalyCLR representations compared to raw data in an anomaly detection task we use the CMS anomaly detection challenge dataset [66], which contains simulated proton-proton collisions with a 13 TeV centre-of-mass energy. The events

are selected to have at least one $e$ or $\mu$ with transverse momenta $p_T > 23$. The pseudo-rapidity ($|\eta|$) is required to be $< 3$ and $< 2.1$ respectively for $e$ and $\mu$. Further, the events are allowed to have up to 10 jets with $p_T > 15$ GeV and $|\eta| < 4$, up to 4 muons $p_T > 3$ GeV and $|\eta| < 2.1$, up to 4 electrons $p_T > 3$ GeV and $|\eta| < 3$ and missing transverse energy (MET). The dataset is generated with Pythia 8.240 generator [67] with a fast detector simulation using Delphes 3.3.2 [68] with the Phase-II CMS detector card. The jets are reconstructed using anti-$k_t$ algorithm [69] in FastJet [70]. In the provided dataset each event is formatted such that the first entry is assigned for MET, next eight are assigned for electrons and muons respectively and, the final 10 entries are for jets. For each particle object the data set contains information of $p_T$, $\eta$, $\phi$ and particle id such that the shape of an event in the data frame is [N,19,4] where N is the total number of events. Note that if an event has less than the maximum allowed of a type of object, the remaining entries in that case are zero padded. The background dataset consists of a number of Standard Model processes and to determine the performance of the anomaly detection algorithm four light BSM scenarios are considered.

## Backgrounds

For the SM background a collection of events are generated from production channels with at least a single lepton in the final state. The fraction of events to be included in the SM for each process is fixed by its trigger efficiency and the LO cross section. Thus, four leading processes are considered: $W$ and $Z$ inclusive productions, QCD multijet contributions, and $t\bar{t}$ production. The proportions between the four processes are given in [71] as:

$$
\begin{aligned}
pp &\rightarrow W^\pm + \text{jets} \rightarrow \ell^\pm \nu_\ell + \text{jets} & (59.2\%) \\
pp &\rightarrow Z + \text{jets} \rightarrow \ell^+ \ell^- + \text{jets} & (6.7\%) \\
pp &\rightarrow t\bar{t} + \text{jets} & (0.3\%) \\
pp &\rightarrow \text{jets} & (33.8\%) \, .
\end{aligned}
\tag{1}
$$

with $\ell = e, \mu, \tau$. The QCD multijet production is by far the largest production process at the LHC. Although leptons in QCD multijet backgrounds are rarely present and mainly originate from decays of unstable hadrons, the sheer volume of QCD multijet production makes it one of the largest processes in the data stream for the challenge.

## New physics signals

The signal datasets provided by the challenge consist of events simulated from the following signal models:

- **Leptoquark (LQ)**: A 80 GeV LQ decaying in to a $b$ and $\tau$.
- **Neutral scalar boson $A$**: A 50 GeV neutral scalar boson $A$. The production mechanism $pp \rightarrow A + X \rightarrow Z^* Z^* + X$ (with $X$ is inclusive activity) followed by both $Z^*$ decaying into charged leptons.
- **Scalar boson $h^0$**: A scalar boson 60 GeV $h^0$ with $pp \rightarrow h^0 + X \rightarrow \tau^+ \tau^- + X$ production.
- **A charged scalar $h^\pm$**: Charged scalar with 60 GeV mass and $pp \rightarrow h^\pm + X \rightarrow \tau \nu + X$ production.

The most distinguishing high-level features of these signals when compared with the background processes are the electron, muon, and jet multiplicities and the $p_T$ and MET distributions [*][†] .

---

[*]We note that since the publication of previous papers using this dataset, a bug fix in the simulation has resulted in a new dataset, and so it is difficult to make direct comparisons between new and old results.

# 3 AnomalyCLR

In this section we describe the AnomalyCLR method [‡]. Contrastive learning of representations (CLR) [52] is a technique used to construct highly-expressive representations of data for use in downstream tasks, in our case this task is anomaly detection. It is self-supervised in that the technique does not require any 'truth' labels for the training data. The advantage of this from the collider physics perspective is that the technique could be run directly on experimental data rather than on simulation. Due to the ability of deep learning methods to learn non-trivial correlations in data that is not expected to be well-modelled by simulation, this is an important aspect of CLR for anomaly detection.

## 3.1 Contrastive learning

The basic idea is that some function $f(\cdot)$ (typically a neural network) is used to map from the data space $\mathcal{D}$ to a representation space $\mathcal{R}$, with the function being optimised to solve some auxiliary task which does not require truth labels. This auxiliary task is framed as an optimisation problem using 'pseudo-labels'. In the anomaly detection scenario addressed in this work, the function that performs the mapping from $\mathcal{D}$ to $\mathcal{R}$ is optimised only on background data. Given that the collider events or objects such as jets typically consist of unordered sets of particles reconstructed by the experiment, we opt for a permutation-invariant function to perform the mapping from $\mathcal{D}$ to $\mathcal{R}$. Specifically, we use a transformer encoder neural network, there are more details on this later in the section.

The auxiliary task that our function is optimised to solve uses augmentations of the collider data. In the traditional contrastive learning approach these augmentations are used to define two types of pseudo-labels:

1. Positive-pair labels
   These labels match each data point in the sample to an augmented version of itself.
2. Negative-pair labels
   These labels match each data point in the sample to every other data point which is not itself or an augmented/transformed version of itself.

The function $f(\cdot)$ is then trained to map from the raw data to the representation space such that positive-pairs are close together in $\mathcal{R}$ and negative-pairs are far apart in $\mathcal{R}$. These two optimisation goals are referred to as alignment (of positive-pairs) and uniformity (of negative-pairs), respectively. The augmentations are chosen to be modifications of the data that should leave the underlying physics unchanged, for example a symmetry in the physical system or an augmentation that could mimic a detector resolution effect.

Each data point is described by an array of data $x_i$ with the subscript labelling the specific data point. We denote an augmentation of a data point as $x_i'$, with the positive-pairs and negative-pairs being defined as the sets $\{(x_i, x_i')\}$ and $\{(x_i, x_j)\} \cup \{(x_i, x_j')\}$ for $i \neq j$, respectively. The contrastive loss function that the network is trained to minimise then is

$$\mathcal{L}_{\text{CLR}} = -\log \frac{e^{s(z_i, z_i')/\tau}}{\sum_{j \neq i \in \text{batch}} \left[ e^{s(z_i, z_j)/\tau} + e^{s(z_i, z_j')/\tau} \right]} , \tag{2}$$

where $z_i = f(x_i)$ and $z_i' = f(x_i')$ are the outputs of the mapping function. The cosine similarity measure $s(\cdot, \cdot)$ is used to compare events and measure distances between them in the new

---

[‡]The code will be made available at https://github.com/bmdillon/AnomalyCLR.

representation space,

$$s(z_i, z_j) = \frac{z_i \cdot z_j}{|z_i||z_j|} = \cos\theta_{ij} \, . \tag{3}$$

In this way, $s(\cdot, \cdot)$ projects each vector $z_i$ to the surface of a unit hypersphere and computes the cosine distance between each pair. As it stands, $s(\cdot, \cdot)$ is not a proper distance metric, however we could form one by taking $d_{ij} = \theta_{ij}/\pi$ as the distance between each event in the representation space, although we do not explore this here. The numerator of the contrastive loss in Eq. (2) accounts for the positive-pair and alignment, where distances between events and their augmented counter-parts enter. While the denominator accounts for the negative-pairs and uniformity, where distances between completely different events are accounted for. The degree to which we trade off between the different tasks is determined by the temperature hyper-parameter $\tau$ in the loss function.

## 3.2   CLR for anomaly detection

While contrastive learning has been shown to be very useful in generating representations for downstream classification tasks [53], there is a potential issue when using this approach for downstream anomaly detection tasks. For the classification task, for example in [53], the function $f(\cdot)$ is optimised on data from both the background and signal classes, despite not using their truth-labels explicitly in the optimisation. Through the contrastive learning this allows the function to encode non-trivial features of both the background and signal data in the representations. When using contrastive learning for a downstream anomaly detection task however, the function $f(\cdot)$ is optimised on just the background data (or at least a significantly background-dominated dataset). This means that the representation learned by the function $f(\cdot)$ will focus solely on features relevant for the background data. This could mean that anomalous data is not out-of-distribution and so may not lead to competitive performance in downstream anomaly detection tasks. This will become evident when we look at the results in Section 6. To remedy this we introduce AnomalyCLR, a modified approach to contrastive learning for anomaly detection in particle physics. At the core of this approach is the introduction of 'anomaly-augmentations', such that we now have two categories for augmentations:

1. Physical augmentations
   These are augmentations of the data that we would like the mapping to be invariant to.
2. Anomaly-augmentations
   These are unphysical augmentations of the data that are supposed to mimic potential anomalies, we want the representations to be highly discriminative towards these augmentations.

We add a third pseudo-label:

3. Anomaly-pair labels
   These labels match each data point in the sample to an anomaly-augmented version of itself.

The advantage of anomaly-augmentations is that we can increase the sensitivity of the anomaly detection tools to anomalies using just the background data, potentially the data directly measured at colliders. This keeps the approach in line with the original data-driven CLR idea. We can then define the anomaly-augmented contrastive loss function as

$$\mathcal{L}_{\text{AnomCLR}} = -\log \frac{e^{\left[s(z_i, z_i') - s(z_i, z_i^*)\right]/\tau}}{\sum_{j \neq i \in \text{batch}} \left[e^{s(z_i, z_j)/\tau} + e^{s(z_i, z_j')/\tau}\right]} \, , \tag{4}$$

where we denote the representations of the anomaly-augmented events by $z^*$, and so the anomaly-pair is defined as $\{(x_i, x_i^*)\}$. Note that the anomaly-augmentations only enter in the numerator of Eq. (4), and without these the loss function becomes the regular contrastive loss function. Introducing the anomaly-pairs we expose the network to data features that are outside of the background distribution. The CLR portion of the loss function still optimises for alignment and uniformity, however this uniformity is now disrupted by the anomaly-pair term. As a result the background data will not be uniformly distributed in the representation space, with some regions encoding features of the anomaly-augmented data. This means that anomalous data with features similar to those generated by the anomaly-augmentations should be out-of-distribution in this representation space.

We did some minor testing on alternative forms of this loss function, for example including the anomaly-augmentations in the denominator of the loss function with the negative-pairs. However since the anomaly-augmentations compute distances between a data point and its augmented counter-part, and not between other data points (i.e. $i \neq j$), it is more intuitive to include this term in the numerator. The denominator in Eq. (4) is used to encode features in the representation space that discriminate between the different data points used during training, which for anomaly detection is the background data. This is not necessary for anomaly detection, and the anomaly-pairs should provide the representations with all the discriminative power they need, so we experimented with removing the denominator in Eq. (4) altogether, and found that this is sufficient. In this case the loss function is written as

$$\mathcal{L}_{\text{AnomCLR}}^+ = -\log e^{\left[s(z_i, z_i') - s(z_i, z_i^*)\right]/\tau} = \frac{s(z_i, z_i^*) - s(z_i, z_i')}{\tau} \, , \tag{5}$$

where the plus sign in $\mathcal{L}_{\text{AnomCLR}}^+$ indicates that only positive-pairs are used. This results in a much less computationally expensive loss function, since we no longer need to compute pairwise correlations between each entry in a batch the complexity scales as $N_{\text{batch}}$ rather than $N_{\text{batch}}^2$. We also remove the dependence on $\tau$ in $\mathcal{L}_{\text{AnomCLR}}^+$, since there is no longer a trade-off between positive- and negative-pairs. We could of course introduce a term to control the trade-off between the physical and anomaly-augmentation terms, but we do not explore that here. In our results we will compare the performance of both loss functions. The number of augmentations is theoretically unlimited, however including a large number of scenarios can incur unstable optimization of the loss especially for contradicting transformations. This problem can be tackled with a larger batch size, to get a better average estimate of the loss, and with a larger representation space.

## 4  Application to event-level anomalies

The application of AnomalyCLR to different physical scenarios requires an understanding of the data and the physics in order to construct the physical and anomaly-augmentations. For the event-level dataset discussed in Section 2 we consider three physical augmentations to the data:

1. Azimuthal rotations
   The whole final state is rotated by an angle $\phi$ randomly sampled from $[0, 2\pi]$.
2. $\eta - \phi$ smearing
   The $(\eta, \phi)$ coordinate of every object in the event is resampled according from a Normal distribution centred on the original coordinate and with a variance inversely proportional to the $p_T$, i.e. $\eta' \sim \mathcal{N}(\eta, \sigma(p_T))$ and $\phi' \sim \mathcal{N}(\phi, \sigma(p_T))$.
3. Energy smearing

The $p_T$ of every object in the event is re-sampled according to $p_T' \sim \mathcal{N}(p_T, f(p_T))$ with $f(p_T)$ determining the strength of the smearing.

These augmentations reflect both the symmetries in the data and the experimental resolution of the detector. Detectors are imperfect, especially in measuring jet energies, and we encode this in the representations of the data through the energy-smearing augmentation. Here we re-sample the jet $p_T$'s as $p_T' \sim \mathcal{N}(p_T, f(p_T))$, where $f(p_T) = \sqrt{0.052 p_T^2 + 1.502 p_T}$ is the energy smearing applied by Delphes (the $p_T$'s are normalised by 1GeV). If not explicitly mentioned, we always assume units of GeV for energy. For the anomaly-augmentation we consider some very simple scenarios:

1. Multiplicity shift, $x_i' = m(x_i)$
   For each event $m(\cdot)$ adds a random number of electrons, muons, and jets to the event. The number is chosen randomly within the limits $(n_e, 4-n_e)$, $(n_\mu, 4-n_\mu)$, and $(n_j, 10-n_j)$ for electrons, muons, and jet, respectively. The azimuthal angle and pseudo-rapidities are also chosen randomly within the limits allowed, and the $p_T$ for each object is chosen as a random fraction of the maximum $p_T$ in the event. Once the objects have been added, the MET of the event is recalculated and updated.

2. Multiplicity shift, keeping MET and the total $p_T$ constant, , $x_i' = \overline{m}(x_i)$
   This is similar to the above augmentation, but now $\overline{m}(\cdot)$ generates the extra objects by splitting the existing objects and smearing the $\eta - \phi$ coordinates using the function used in the physical augmentations above.

3. $p_T$ and MET shifts, $x_i' = s_{p_T}(x_i)$
   Here $s_{p_T}(\cdot)$ shifts the $p_T$'s in the event by the same random factor. We randomly choose whether we shift just the MET, just the reconstructed object $p_T$'s, or both. And we ensure that the the trigger selection is not spoiled by these shifts.

With the physical augmentations we apply all of them simultaneously, whereas for the anomaly-augmentations we apply just one augmentation to each event. The augmentation that is applied is selected randomly and uniformly. We do not apply both a physical augmentation and an anomaly-augmentation to the events in $s(z_i, z_i^*)$, since this would conflict with the optimisation goal of the $s(z_i, z_i')$ term. It would also be possible to have an anomaly-augmentation that removes objects from the event, however this effect is already captured by the augmentation that adds objects to the event. Many of the events in the background have the minimal multiplicity allowed by the applied cuts, so the effect of an anomaly-pair with a low multiplicity background event and the same event augmented to have more objects is the exact same as the effect of an anomaly-pair with a high-multiplicity background event augmented to have less objects. This is because of the symmetry in the distance function $s(z_i, z_i^*)$. So the anomaly-augmentations here are as general as can be, and do not target any specific new physics scenario, therefore the technique should be model-agnostic. More precisely, the anomalous transformations democratically introduce a modification given the background data and its format. In our implementation there are no explicit assumptions on the allowed signals, both multiplicity shifts, $m(\cdot)$ and $\bar{m}(\cdot)$, uniformly sample the number of additional reconstructed objects and differ in the treatment of the kinematics. In the first case, the new objects take a fraction, uniformly sampled as well, of the maximum $p_T$, while in the second case the total MET and the $p_T$ are left constant. The third augmentation shifts MET and/or $p_T$ uniformly sampling a scaling factor, given the original values. Here, we fixed a window corresponding to five times the original momentum. This window has not been fine tuned and, given the wide range of kinematics in the training events, this window covers an extremely large phase-space region. Given the generality of these transformations, the representations comply with our anomaly detection downstream task. For the general application of this method, it is important a careful study of the augmentation technique and their implementations to avoid the

usage of inconsistent data.

**Architecture and training**

The collider event data being used has a well-defined structure:

- MET: one entry with $(p_T, \eta, \phi)$
- Electrons: four entries, each with $(p_T, \eta, \phi)$
- Muons: four entries, each with $(p_T, \eta, \phi)$
- Jets: ten entries, each with $(p_T, \eta, \phi)$.

This amounts to a $19 \times 3$ array, with the electrons, muons, and jets being ordered by $p_T$ and having zero-padded entries where there is less than the maximum allowed number of reconstructed objects. The multiplicity is typically much less than the maximum allowed, so the data for a single collider event can have many zeros. The transformer allows us to avoid this by having a permutation-invariant and variable length input format. Because the data is now processed in a permutation-invariant way, the information on which entry corresponds to which object (MET, electron, muon, or jet) is lost. We reinstate this information by adding a one-hot encoded ID vector to $(p_T, \eta, \phi)$, with a 1 indicating the correct ID. This means that each reconstructed object is now represented by a 7D vector. Before passing the kinematic data to the transformer we do some very minor preprocessing to make sure that the numbers the networks see are $\mathcal{O}(1)$. Specifically, we divide all MET and $p_T$ values by the average $p_T$ of all objects (electrons, muons, jets) in the background dataset, we do not shift the values to be centred on zero because the distribution is highly peaked at zero and we want the preprocessed data to have the same sparsity as the original data. We then divide all $\eta$ and $\phi$ values by 4 and $\pi$, respectively. When training the AutoEncoder networks discussed in the next section we use the same preprocessing of the data, this ensures that any difference in the results can be attributed to AnomalyCLR.

The transformer starts by projecting each object to a larger vector whose dimension is determined by the embedding dimension. The embeddings for each object are then passed through the transformer, with a feed-forward network between each transformer layer. The output from the transformer has a dimension of ($n \times$ model dimension) with $n$ being the number of objects in the event. The last steps are to sum over the $n$ vectors in this output, which enforces the permutation-invariance, and to pass this vector through a fully-connected head network. The output of this head network is what is passed to the loss function. For more details on the architecture we refer the reader to [53], here we only list the hyper-parameters used in training the network in Table 1. The representation used in the anomaly detection task is taken from the output of the transformer network, before being passed through the head network. It is well documented in the machine learning literature that these intermediate representations from self-supervised networks generally contain more discriminating features, for example in [52].

## 5 Anomaly scores

The basic flow in an AutoEncoder involves two steps; (i) mapping high-dimensional input data to a compressed latent space using a neural network called an encoder, and (ii) mapping the compressed latent space representation to a reconstructed version of the input data using a neural network called a decoder. We refer to the encoder network as $e(\cdot)$ and the decoder network as $d(\cdot)$. With input data of dimension $D$, and a bottleneck of dimension $B$, the encoder maps $e : \mathbb{R}^D \rightarrow \mathbb{R}^B$, while the decoder maps $d : \mathbb{R}^B \rightarrow \mathbb{R}^D$, with the AutoEncoder defined as

| hyper-parameter | | hyper-parameter | |
|---|---|---|---|
| model (embedding) dimension | 160 | optimiser | Adam($\beta_1$=0.9, $\beta_2$=0.999) |
| feed-forward hidden dimension | 160 | learning rate | $5 \times 10^{-5}$ |
| output dimension | 160 | batch size | 128 |
| # self-attention heads | 4 | # epochs | 300 |
| # transformer layers ($N$) | 4 | | |
| # layers | 2 | | |
| dropout rate | 0.1 | | |

Table 1: Default setup of the transformer-encoder network and the AnomalyCLR training, unless noted explicitly.

$h = e \circ d : \mathbb{R}^D \to \mathbb{R}^D$. Acting on a single input $\vec{x}$, the AutoEncoder returns $\vec{x}' = h(\vec{x})$, and is optimised to minimise the mean-squared-error (MSE) loss function between the input and reconstructed input,

$$\mathcal{L}(\vec{x}, \theta) = \left( \vec{x} - \vec{x}' \right)^2 \, , \tag{6}$$

where $\theta$ represents the learnable parameters of the AutoEncoder. In the limit where the AutoEncoder is able to reconstruct inputs perfectly, which is guaranteed to be possible when $D = B$, the function $h_\theta(\cdot)$ is simply the identity. But with $B < D$ the AutoEncoder may not be able to perfectly reconstruct all features in the data, and therefore it should learn to reconstruct only the most common or prominent features in the data. This means that events containing rare or anomalous features should have a larger 'reconstruction loss', i.e. $L(\vec{x}, \theta)$, and this can then be used as the anomaly score.

The encoder and decoder networks have 5 feed forward layers each with 256, 128, 64, 32, and 16 neurons, connected by a 5-dimensional bottleneck. The activation function between layers is a LeakyReLU with default slope. The decoder is a mirrored version of the encoder. We don't apply regularization techniques during training. The training is performed using Adam optimiser with learning rate 0.001 for 100 epochs, the batch size is 4096, and the number of SM events used is $10^6$. Note that we have not optimised the AutoEncoder architecture, simply choosing the same architecture used in [39]. Instead we have only ensured that they are trained to convergence and that the training is stable. The AutoEncoder is trained on both the representations obtained from contrastive learning and the raw data. In the case of the raw data we apply the same preprocessing to the data as is applied to the data in the contrastive learning network. In this way we ensure that any differences in the anomaly detection performance can be attributed to the contrastive learning methods.

# 6 Results

In this section we present some results using the different techniques discussed in the preceding sections. The results here are three-fold; we first compare the different methods based on anomaly detection performance, we then study the effects of the different anomaly-augmentations on the AnomalyCLR performance, and lastly we look at the effect of the representation dimension on the performance.

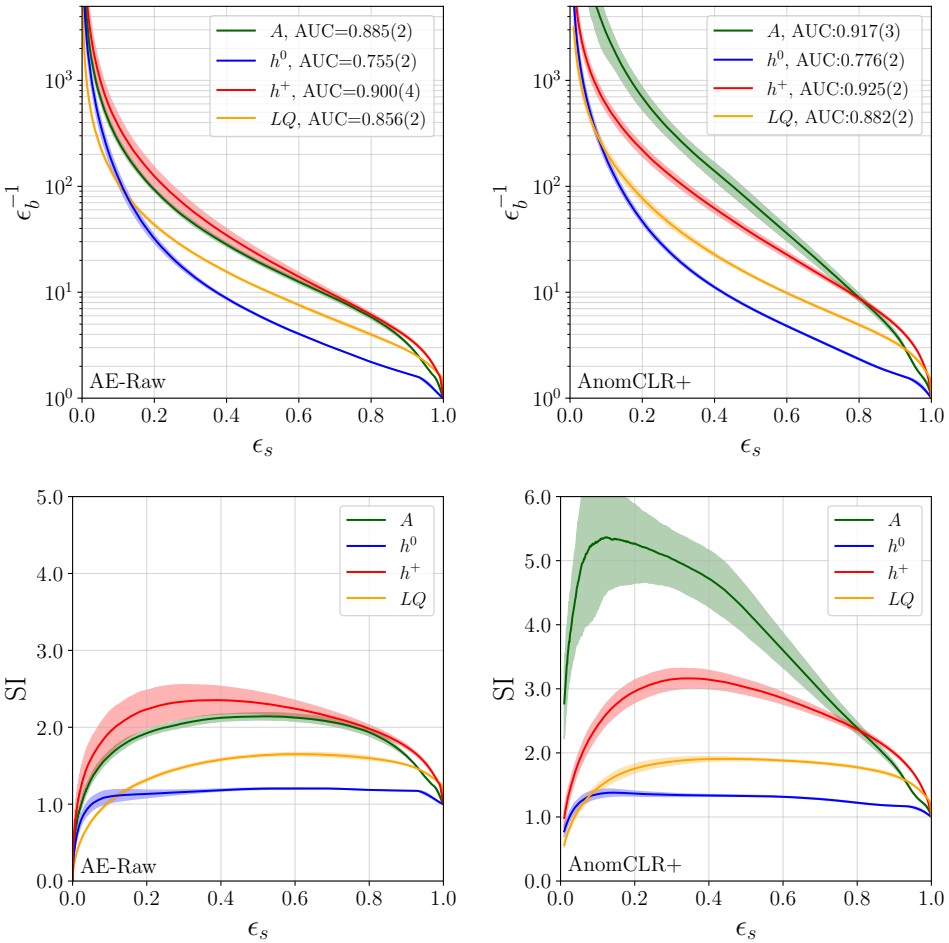

Figure 1: Comparison between the AE on raw data and the AE on the CLR representations trained with the $\mathcal{L}_{\text{AnomCLR}}^+$ loss function.

## 6.1 Comparison of methods

We compare the methods using the ROC (Receiver Operating Characteristic) curves, the SI (Significance Improvement) curves, and the AUC (Area Under the ROC Curve). The baseline we compare to is the AutoEncoder trained on raw kinematic data. We present results using the CLR method without anomaly-augmentations ($\mathcal{L}_{\text{CLR}}$), and the CLR method with anomaly-augmentations (both $\mathcal{L}_{\text{AnomCLR}}$ and $\mathcal{L}_{\text{AnomCLR}}^+$). So we have 4 methods in total to compare. For all results on the raw data we have trained five AutoEncoder networks and taken the central value and the error estimation from the mean and standard deviation of the results. For the CLR methods we also aggregate over five different CLR runs, where for each run we train a different transformer network and three different AutoEncoders. We then take the central value and the standard deviation for the error estimate. The CLR representations have a dimension of 160 and where anomaly-augmentations are used we have used them all as outlined in Section 4. In Fig. 1 we present AnomalyCLR results using $\mathcal{L}_{\text{AnomCLR}}^+$ and see that it leads to significant improvements over the raw data representations, not only in the AUC but also at all signal efficiencies. In the Significance Improvement (SI) curves we also see large improvements, with the SI being between $\sim 3.5$–$4$ for $A \rightarrow 4l$ and $h^+$. We can see from Table 2 that the $\mathcal{L}_{\text{AnomCLR}}^+$ loss function is clearly advantageous over $\mathcal{L}_{\text{AnomCLR}}$, beating it on all signals with the exception of $A \rightarrow 4l$, where $\mathcal{L}_{\text{AnomCLR}}$ achieves better performance at $\epsilon_s = 0.3$. A point of interest here is that the AutoEncoder on raw data outperforms the AutoEncoder on the CLR

representations in most cases. This is likely due to the fact that traditional CLR optimises for uniformity, and since it is trained on background only, the mapping is not optimised to separate SM-like background events from any event which may look different to that. The benefit of anomaly-augmentations here is strikingly clear.

## 6.2 The effect of anomaly-augmentations

We now want to study how the addition of the individual anomaly-augmentations affects the anomaly detection performance. For this we use just $\mathcal{L}^+_{\text{AnomCLR}}$ , however we expect the results with $\mathcal{L}_{\text{AnomCLR}}$ to be similar. We use a representation dimension of 160 and obtain the error estimate on the runs with a combination of different CLR and AutoEncoder trainings. We train five different CLR models, and then train three separate AutoEncoders on each of these models, and take the average and standard deviation to obtain the error. We can see from Fig. 2 that the affect of the augmentations together results in more or less the best overall performance. One thing we noticed is that it can be difficult to determine from the affect of individual augmentations, or subgroups of them, what the performance of all of them together will be. For example, in most cases if we take just the $m(x)$ augmentation, i.e. the multiplicity augmentation that simply adds reconstructed objects, we see that it alone decreases performance below baseline for two out of four signals. However when used in combination with the others it either increases or has little effect on the performance. We can in fact see from this figure which augmentations are most advantageous for each signal. For the $LQ$ this is the $m(x)$ augmentation, for $h^0$ it is both the $m(x)$ and $s_{p_T}$ augmentations, for $A$ it is the $\overline{m}(x)$ augmentation, and for $h^+$ it is the $s_{p_T}$ augmentation. The important take away here is that in the case where we do not know what the signal is, including all augmentations will allow us to be signal-agnostic and retain the discriminative power for each signal type.

## 6.3 The effect of representation dimension

With CLR we can project our raw data from $\mathcal{D}$ to a representation of any dimension we like. We would expect that the larger the representation dimension the more information that can be encoded in the space. However we also expect that this would plateau or even peak at some point, and this what we want to investigate here. For this we use just $\mathcal{L}^+_{\text{AnomCLR}}$ , however we

|  | Signal | AE-Raw | CLR | AnomCLR | AnomCLR+ |
|---|---|---|---|---|---|
| AUC | $A$ | 0.885(2) | 0.89(1) | **0.918(2)** | **0.917(3)** |
|  | $h^0$ | 0.755(2) | 0.726(9) | 0.749(3) | **0.776(2)** |
|  | $h^+$ | 0.900(4) | 0.84(1) | 0.898(3) | **0.925(2)** |
|  | $LQ$ | 0.856(2) | 0.82(1) | 0.847(6) | **0.882(2)** |
| $\epsilon_b^{-1}(\epsilon_s=0.3)$ | $A$ | 47(2) | 170(70) | **400(100)** | **270(50)** |
|  | $h^0$ | 14.9(7) | 10(1) | 15.0(5) | **19.1(7)** |
|  | $h^+$ | 60(10) | 20(2) | 53(3) | **110(10)** |
|  | $LQ$ | 24.4(6) | 16(1) | 27(1) | **37(2)** |
| SI($\epsilon_s=0.3$) | $A$ | 2.05(5) | 4.0(8) | **6.4(9)** | **5.0(4)** |
|  | $h^0$ | 1.16(3) | 1.01(5) | 1.19(2) | **1.34(2)** |
|  | $h^+$ | 2.3(2) | 1.38(9) | 2.24(7) | **3.2(2)** |
|  | $LQ$ | 1.48(2) | 1.25(5) | 1.59(3) | **1.87(6)** |

Table 2: Comparison of the different CLR loss functions, with and without anomaly-augmentations, and the AE trained on raw data.

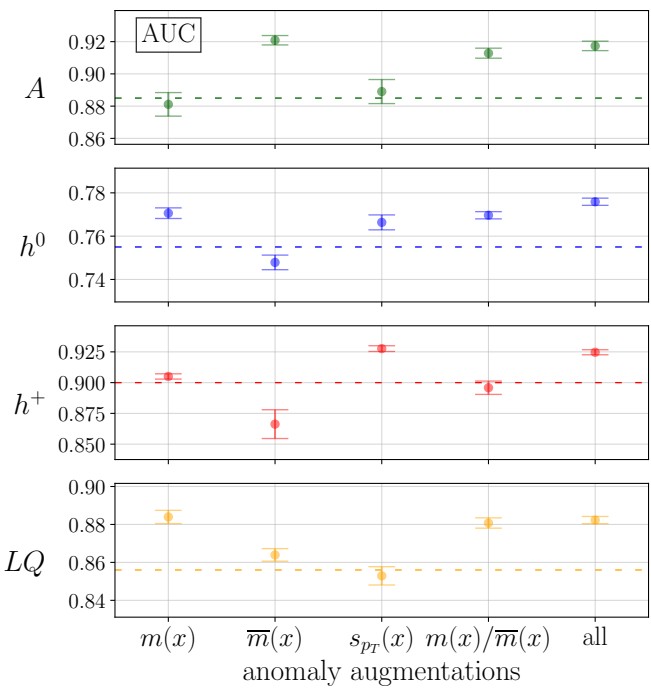

Figure 2: Results of a scan on the anomaly-augmentations used with the $\mathcal{L}^+_{\text{AnomCLR}}$ loss function. The augmentations are defined in Section 4. The dashed lines here correspond to the AutoEncoder on raw data baseline performance.

expect the results with $\mathcal{L}_{\text{AnomCLR}}$ to be similar. Here we also obtain the error estimate from a combination of five CLR models and three AutoEncoders trained on each.

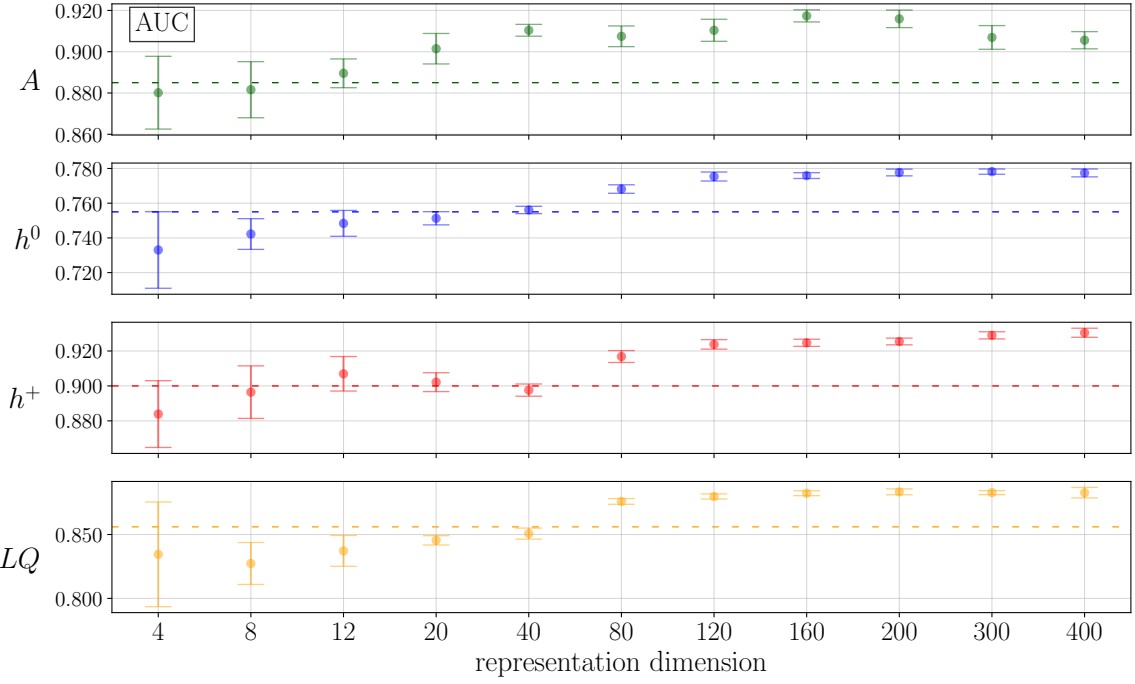

Figure 3: Results of a scan on the representation dimension used with the $\mathcal{L}^+_{\text{AnomCLR}}$ loss function. The dashed lines here correspond to the AutoEncoder on raw data baseline performance.

In Fig. 3 we see that increasing the representation dimension certainly improves the performance of the anomaly detection, at least up until a certain point. The $A$ signal appears to achieve peak performance somewhere between dimensions 120 and 200, while the other signals performances all increase and/or plateau right up until 400. There is no fundamental limitation related to the representation size which we would expect to cause a degradation at larger dimensions, however there are two points we should keep in mind here. The first is simple, these means and variances are calculated with a total of fifteen runs (five CLR models each with three AutoEncoders), so more runs might present a clearer picture. The second point is that we have not optimised the AutoEncoder architecture or hyper-parameters as the representation size increases. While it is beyond the scope of this paper, it is possible that an independent hyper-parameter optimisation for each representation dimension would improve these results, particularly at larger dimensions. What these results show is that there is a clear tendancy for the results to improve as we increase from dimensions of $\sim 4$ to $\sim 100$, as we would naturally expect.

## 7    Summary & conclusions

In this paper we have introduced AnomalyCLR[§], a new method for density-based anomaly detection in high-energy physics. It makes use of anomalous augmentations of collider data to build a representation space from which to construct anomaly scores with a range of methods, for example using AutoEncoders. It is a self-supervised method, based on the contrastive learning idea. We tested this method on the CMS ADC dataset, and compared to the raw data baselines we find large improvements on all signals. At a fixed signal efficiency of 0.3 and a fixed representation dimension of 160 we find significance improvements for the different signals in the range of 14–70%, and a decreased relative error on the significance improvement in each case. Allowing for varying signal efficiencies and representation dimensions would improve these performance markers even further.

Density-based anomaly detection, using AutoEncoders or normalising flows, suffer from the ambiguity that a change in the 'coordinate system' or representation of the data results in a fundamental change in how the anomaly score is defined. This makes it difficult to choose a suitable representation by hand, for example a simple re-mapping of $p_T$'s along with some re-scaling of numerical inputs. These simple choices are difficult to motivate from a physics perspective and can drastically change the results of the anomaly detection. This change can be for better or for worse, and typically depends on the signal models used to test the algorithm.

AnomalyCLR addresses this by constructing a representation of the data using self-supervised contrastive learning with the addition of anomaly-augmented data. The anomaly-augmented data is constructed from the background data through feature augmentation, designed to emulate a generic anomaly. We have discussed in detail how we do this for the event-level anomalies in the CMS ADC dataset, however this would of course be different in different physics cases. We proposed a new loss function which we use to train a deep transformer-based neural network. This network projects the events to a new representation, in which the anomaly-augmented events are far from their original counterparts, while being close to events which are similar. The transformer network then learns a highly discriminative representation of the events which is sensitive to the presence of potential anomalies. We have seen that the choice of these augmentations is quite model agnostic. This model-agnostic nature of the approach can be seen in how the results improve across all four signals considered.

---

[§]The AnomalyCLR code, along with the event-level anomaly detection application, are made available at https://github.com/bmdillon/AnomalyCLR.

We have shown the effectiveness of self-supervision and the idea of anomaly-augmentations in significantly enhancing anomaly detection performance in a model-agnostic way. This opens the door to further studies, such as improving the density-estimation portion of the method with a more sophisticated hyper-parameter optimisation of the AutoEncoders, using normalising flows, or even using the Normalised AutoEncoder. More generally, the use of anomaly-augmented data could be explored further in other anomaly detection approaches.

## Acknowledgements

We would like to thank Jernej Kamenik and Ben Nachman for their helpful comments on the manuscript. LF would like to thank Jessica N. Howard for useful comments on the implementation of the different augmentations. BMD acknowledges funding from the Alexander von Humboldt Foundation. LF, TM, and TP are funded by the Deutsche Forschungsgemeinschaft (DFG, German Research Foundation) under grant 396021762 – TRR 257: Particle Physics Phenomenology after the Higgs Discovery and Germany's Excellence Strategy EXC 2181/1 - 390900948 (the Heidelberg STRUCTURES Excellence Cluster).

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
