# Peer review of "Anomalies, Representations, and Self-Supervision"

_SciPost Physics_

## Round 1 · Referee Report · Anonymous (Referee 1) · 2023-5-9

Report

The authors present a framework for anomaly detection using contrastive learning. The paper is well written and provides a contribution to the ongoing ML efforts in particle physics. Therefore, I am happy to recommend publication in SciPost after the authors address the following point:

The presented data augmentations for anomaly modelling are not exhaustive. The anomaly signatures that they discuss would also be hard to miss in a more traditional approach to BSM discrimination. Are there issues of the framework with respect to scalability, would adding an order of magnitude of scenarios in parallel change the ability for classification. Also, do the authors believe that this procedure would be indeed perform superior in comparison to GANs pivoting background uncertainties?

Additional minor point:

three physical augmentations the data: -> to the data?
  • validity: -
  • significance: -
  • originality: -
  • clarity: -
  • formatting: -
  • grammar: -

Author:  Barry Dillon  on 2024-06-06  [id 4541]

(in reply to Report 1 on 2023-05-09)

We would like to thank the referee for their patience in receiving these updates.

Since our approach is developed for anomaly detection rather than classification, so it can be difficult to compare to other ML classifiers or traditional BSM searches where the signal features are precisely specified, although in Sec.3.2 we compare the two points of view. Even a pivot classifier that can reduce the dependence on nuisance parameters will optimize for the likelihood ratio of the given two samples, introducing a dependence on the particular signal.

Generally, the addition of more augmentations will reduce the similarity of either positive or anomalous pairs. However an increase of the latent/representation dimension can mitigate this issue by introducing more space where events can be organized. Our selection is based on general features of possible BSM signals and the framework allows for any number of augmentations. To avoid signal dependence, it is important to use a broad variety of augmentations. We added a comment on this in Sec.3.2.

---

## Round 1 · Referee Report · Anonymous (Referee 2) · 2023-10-12

Report

I appreciate your patience in receiving this feedback. In this paper, the authors delve into the application of contrastive learning for anomaly detection in the field of High-Energy Physics (HEP). The manuscript is well-crafted, and I have only a few minor comments to add:

  1. In Section 2: Delphes utilizes FastJet in the backend. However, the authors haven't provided a citation for this tool; only the anti-kT algorithm is referenced.

  2. In Section 3.2: It would be beneficial if the authors could provide a more detailed explanation of anomaly augmentations. The current text might be misconstrued as suggesting that these augmentations are model-dependent, influenced by the choice of augmentation technique. Clarity on this aspect would be helpful.

  3. As a general comment, it would be valuable for the authors to discuss the potential reusability of their proposed method beyond simply distributing the learned model. The authors highlight the applicability of their approach to experimental data, which is of great significance. However, there is no mention of how this methodology can be made publicly available for future Beyond Standard Model (BSM) inference beyond the scope of the experimental collaboration.

  4. What kind of metadata should be made available to define the boundaries of the features?

  5. What are the critical nuances that future users should consider to avoid potential extrapolation of the defined model?

These considerations would contribute to the practicality and broader adoption of the AnomalyCLR technique in the HEP community.

  • validity: -
  • significance: -
  • originality: -
  • clarity: -
  • formatting: -
  • grammar: -

Author:  Barry Dillon  on 2024-06-06  [id 4540]

(in reply to Report 2 on 2023-10-12)

We thank the referee for their patience in receiving these updates.

1 - Added FastJet citation.

2 - The referee is correct that the current augmentations do not assume an underlying physics model, with model independence coming from the use of a range of augmentations. We added a paragraph to clarify this aspect in Sec.4, after the description of the anomaly-augmentations.

3 - We provide the codebase to train the model in an open format that includes more than just the trained model, this includes: implemented loss functions, modular base transformer class, and implementation of the augmentations used during training. This provides all the tools to train a model, on any given data, with complete freedom over the choice of augmentations.

Regarding the metadata and crucial nuances mentioned by the referee, we would like to point out that an important aspect of constructing the augmentations is the selection criteria for the events in the dataset. We address this in the paper by constraining the augmented events to be in the acceptance region. This should be addressed case by case for different applications of the AnomalyCLR technique. We had partially overlooked this aspect in the original draft, but have since updated the results and discussion. The results and conclusions have not qualitatively changed. We were also more careful in this analysis, including more CLR and AutoEncoder runs to minimise the uncertainty in the results.

---

## Editorial Decision

resubmitted